# UniAero: A Unified Framework for Global Drag and Local Flow Field Prediction

## Abstract

Automotive aerodynamic design relies simultaneously on global metrics (e.g., drag coefficient $C_d$) and local flow field information (e.g., surface pressure and wall shear stress). While the former dictates overall fuel efficiency, the latter informs detailed design and performance optimization. Existing deep-learning surrogates typically focus on either global or local predictions individually, failing to optimize both tasks jointly, thus limiting their effectiveness in iterative design processes. To address this, we propose **UniAero**, a unified framework that jointly predicts $C_d$ and dense surface fields from 3D automotive geometry. UniAero combines (i) a *Physically Stratified Mixture-of-Experts* (Phys$^2$MoE) with scale-sensitive experts and physics-aware gating for multi-task, multi-scale learning; (ii) *Serialized Patch Attention* to process large meshes efficiently while preserving long-range interactions; and (iii) a hierarchical encoder with *Geometry-aware Position Encoding* (GPE) to capture subtle shape cues. Experiments on three industrial datasets demonstrate that, by explicitly leveraging the inherent coupling between global and local aerodynamic phenomena through joint modeling. UniAero reduces drag error by **12%** and improves local-field accuracy by **16%** over strong single-task baselines, with ~1 s inference per vehicle on a single GPU, far faster than CFD simulations. With its superior accuracy, speed, and coherence, UniAero holds significant promise for automotive aerodynamic design. The code is available at `https://anonymous.4open.science/r/ICLR2026UniAero`.

## 1 Introduction

Automotive aerodynamics significantly impacts vehicle fuel efficiency, driving stability, and overall design quality, making it a critical optimization task in automotive engineering. Traditionally, high-fidelity computational fluid dynamics (CFD) simulations are employed to predict global metrics such as drag coefficient $C_d$, as well as local flow fields like surface pressure and wall shear stress. However, CFD simulations require extensive computational resources, often running for hours or even days on large computing clusters, severely limiting rapid design iterations.

Effective aerodynamic design demands rapid and accurate predictions of both global and local aerodynamic properties. Global metrics (e.g., $C_d$) influence overall vehicle efficiency, while local flow fields provide insights essential for precise geometric refinements. Current surrogate modeling methods predominantly focus either on global aerodynamic metrics (Liu & Chen, 2025; Elrefaie et al., 2024b; Song et al., 2023) or detailed local flow field reconstructions (Wu et al., 2024a; Hassan et al., 2024; Liu et al., 2025; Hao et al., 2023). Models like DrivAerNet++ (Elrefaie et al., 2024b) excel in quick global drag estimation but lack detailed local geometric insights, whereas local-flow-oriented models such as Transolver (Wu et al., 2024a) neglect the fundamental physical coupling between local flow dynamics and global performance metrics. This gap highlights the urgent need for surrogate models that comprehensively address the multi-scale, multi-task nature of aerodynamic flows. Crucially, global aerodynamic metrics and local flow fields are intrinsically coupled, local surface changes impact global drag, and vice versa. Independent modeling neglects these interdependencies, underscoring the need for a unified joint modeling framework.

As illustrated in Figure 1, automotive aerodynamic phenomena inherently exhibit complex multi-scale interactions. Drag primarily depends on large-scale flow structures (e.g., wakes and vortices), while local aerodynamic behaviors (pressure, wall shear stress) are highly sensitive to millimeter-

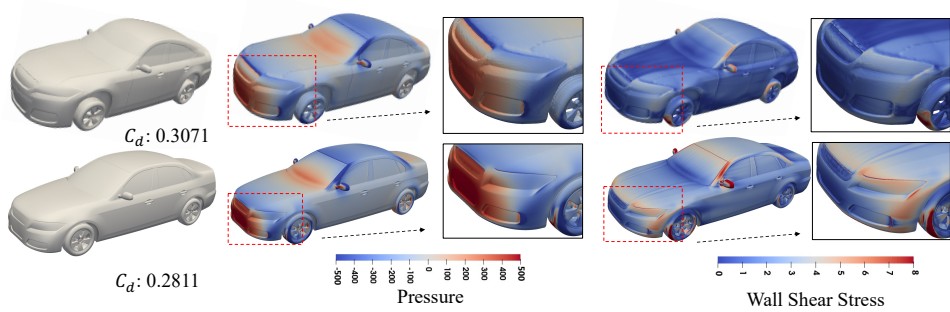

$C_d$: 0.3071

$C_d$: 0.2811

Pressure

Wall Shear Stress

Figure 1: **Global–local aerodynamics of two vehicles with different drag coefficients.** Left: geometry with $C_d$ (top 0.3071, bottom 0.2811) as a global efficiency indicator. Middle/right: surface pressure (P) and wall shear magnitude (WSS) capture fine-grained flow interactions; insets highlight stagnation/separation-prone regions (A-pillar, front wheel arch, grille). Despite the small gap in $C_d$ (scalar), P (signed, wide range) and WSS (positive) differ markedly and non-uniformly. The three targets also span distinct types and scales, spatially heterogeneous differences, making joint learning a cross-scale, heteroscedastic task. Shared color scales; fixed camera/lighting.

scale geometric details. Capturing these cross-scale phenomena poses substantial challenges due to varying loss scales, geometric diversity across vehicle types (e.g., sedans, SUVs), and multi-scale physical interactions. Traditional deep learning models, such as Graph Neural Networks (GNNs) (Deng & Hooi, 2021; Pham et al., 2025) and general-purpose Transformers (Vaswani et al., 2017; Chen et al., 2024; Wu et al., 2022; Liu & Chen, 2025), often struggle to simultaneously capture these intricate dependencies.

To overcome these challenges, we introduce UniAero, a physics-inspired Transformer framework designed explicitly for automotive aerodynamics. UniAero simultaneously predicts global aerodynamic metrics ($C_d$) and local flow fields (pressure, wall shear stress) within a unified and computationally efficient pipeline. The core innovations include: (1) Phys$^2$MoE (Physically Stratified Mixture-of-Experts), employing scale-sensitive expert modules, stratified parameter sharing, and physics-aware routing mechanisms to efficiently handle multi-scale and multi-task complexities in aerodynamic flows; (2) Serialized Patch Attention (SPA), enabling efficient processing of large-scale automotive surface meshes while preserving crucial long-range aerodynamic interactions; (3) Hierarchical Multi-Scale Encoder with Geometry-aware Position Encoding (GPE). Captures global-to-local aerodynamic interactions through a multi-scale encoding structure and explicitly encodes fine-grained geometric information via sparse voxel convolutions, significantly enhancing model sensitivity to subtle surface variations.

Experiments on three industrial-scale automotive datasets (DrivAerNet, DrivAerNet++, and DrivAerML) demonstrate that UniAero substantially outperforms state-of-the-art surrogate models, reducing drag prediction errors by **12%**, improving local flow accuracy by **16%**, and achieving inference times of one seconds per vehicle on a single GPU, orders of magnitude faster than traditional CFD simulations.

In summary, our paper makes three key contributions: (1) We present **UniAero**, a unified Transformer-based surrogate that jointly predicts global aerodynamic $C_d$ and dense local flow fields on automotive geometries; (2) we introduce **Phys$^2$MoE**, a physics-stratified mixture-of-experts that handles multi-scale phenomena and mitigates multi-task conflicts; (3) we design a **hierarchical multi-scale encoder** with geometry-aware positional encoding (GPE) that captures global context and fine geometric cues.

## 2 RELATED WORK

**Surrogate Models in Automotive Aerodynamics.** Effective aerodynamic analysis of vehicles necessitates the accurate prediction of global parameters such as drag coefficient ($C_d$), as well as detailed local aerodynamic characteristics including pressure distributions and wall shear stress. Conventional methods, including computational fluid dynamics (CFD) simulations and wind tunnel experiments (Scardovelli & Zaleski, 1999; Menter et al., 2003; Fröhlich & Von Terzi, 2008), offer

precise results but are computationally expensive and slow, restricting iterative vehicle design processes. Recently, surrogate models leveraging machine learning techniques (Elrefaie et al., 2024b; Liu & Chen, 2025) have been proposed to expedite aerodynamic evaluations. Nevertheless, most existing surrogate methods adopt simplified geometrical representations, such as low-dimensional parameterizations or planar projections, thus inadequately capturing intricate three-dimensional geometric features essential for accurate predictions. UniAero addresses these limitations by integrating comprehensive 3D geometric information into a unified, scalable framework, significantly improving prediction accuracy and computational efficiency.

**Neural Networks for Solving PDEs.** Physics-informed neural networks (PINNs) (Raissi et al., 2019) and neural PDE solvers (Wang et al., 2023; Wu et al., 2024a) have gained prominence in efficiently solving complex fluid dynamics problems governed by partial differential equations. Techniques such as Fourier neural operators (Li et al., 2021; 2023b; Wu et al., 2023a; Li et al., 2023d; Liu et al., 2025) and transformer-based approaches (Liu et al., 2022; Li et al., 2023a; Hao et al., 2023; Wu et al., 2024a; Li et al., 2025b) demonstrate impressive capability in capturing fine-grained local flow phenomena. However, their applicability to global aerodynamic metrics (e.g., drag coefficient) remains indirect and often requires cumbersome post-processing of local solutions, potentially introducing errors and inefficiencies. UniAero circumvents these issues by directly learning a holistic mapping from the vehicle geometry to global aerodynamic indicators, simultaneously incorporating local aerodynamic insights, thereby achieving superior accuracy and streamlined computation.

**Advancements in 3D Geometric Learning.** Recent advancements in geometric deep learning have led to various representations and techniques for analyzing 3D data, including point clouds, voxel grids, and mesh-based methods. Point cloud methods such as PointNet (Qi et al., 2017a) and its derivatives (Pang et al., 2022; Han et al., 2024; Chen et al., 2024) effectively process sparse and irregular geometries, yet encounter limitations with dense and heterogeneous data distributions. Voxel-based methods leverage regular grid structures and convolutional operations (Wu et al., 2024b; 2022; Wang, 2023) but are hindered by significant memory usage and limited spatial resolution. Mesh-based approaches (Pfaff et al., 2021b; Wang et al., 2019) utilize connectivity information but are vulnerable to mesh irregularities and inefficient at handling multi-scale features. Current transformer models, although promising, typically lack explicit integration of critical geometric details specific to automotive aerodynamic analysis. In contrast, UniAero innovatively incorporates hierarchical multi-scale encoding and Geometry-aware Position Encoding (GPE) into transformer architectures, effectively capturing both global interactions and subtle local geometric variations.

**Expert-based Architectures.** Mixture-of-Experts (MoE) models, featuring dynamic routing across specialized subnetworks, have shown considerable promise in multi-task learning and scalable model training (Li et al., 2025a; Wu et al., 2024c; Shazeer et al., 2017). Traditional MoE frameworks primarily focus on improving generalization through sparse gating mechanisms, yet frequently overlook the complexities arising from distinct scale and physics characteristics inherent in specific application domains, such as automotive aerodynamics. UniAero introduces a novel Physically Stratified Mixture-of-Experts (Phys$^2$MoE) architecture that explicitly addresses these challenges by incorporating scale-sensitive experts, stratified parameter sharing schemes, and physics-aware routing mechanisms. This tailored design significantly enhances predictive capabilities in multi-scale, multi-task aerodynamic applications.

## 3 METHODOLOGY

We introduce UniAero, a Transformer-based framework for predicting both global aerodynamic metrics (e.g., drag coefficient $C_d$) and local flow fields (e.g., pressure, wall shear stress) from automotive geometries. Figure 2 shows the architecture.

### 3.1 PROBLEM DEFINITION

Let the vehicle surface be sampled as a point set (or mesh vertices) $\mathbf{G} = \{\mathbf{x}_i\}_{i=1}^N \subset \mathbb{R}^3$. Our goal is to learn a function

$$f : \mathbf{G} \longmapsto \big( \mathcal{F}_{\text{local}}, C_d \big), \tag{1}$$

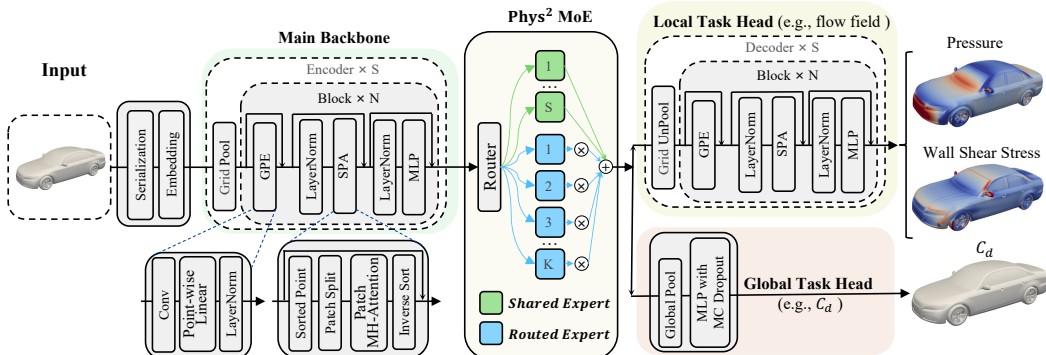

Figure 2: **Overview of UniAero architecture.** The input automotive surface point cloud is serialized and embedded, then passed through a hierarchical Transformer backbone featuring grid pooling, Geometry-aware Position Encoding (GPE), and Serialized Patch Attention (SPA). The extracted multi-scale features enter a Physically Stratified Mixture of Experts (Phys$^2$MoE) module, dynamically routing tokens to shared and task-specific experts. Finally, the local task head reconstructs detailed surface pressure and wall shear stress fields, while the global task head directly predicts the vehicle's drag coefficient ($C_d$).

where $\mathcal{F}_{\text{local}} = \{\mathbf{f}_i\}_{i=1}^{N}$ contains per-vertex physical quantities (e.g. pressure, wall shear magnitude) and global $C_d \in \mathbb{R}$. Equation equation 1 couples local flow reconstruction and global drag regression in a single forward pass, aligning with real-world aerodynamic design needs.

## 3.2 POINT-CLOUD SERIALIZATION

**Motivation.** Aerodynamic simulations of automotive surfaces often involve extremely large point clouds comprising millions of mesh vertices. Accurately modeling airflow dynamics across such extensive surfaces requires capturing intricate local aerodynamic phenomena (e.g., pressure variations, wall shear stress patterns) as well as long-range global interactions influencing overall drag performance. However, traditional attention mechanisms, which rely heavily on computationally intensive K-nearest neighbor (KNN) queries and pairwise interactions, become impractical at such scales. To efficiently handle these computational challenges, we propose a point-cloud serialization technique that transforms the unstructured 3D surface points into a structured, one-dimensional sequence. This allows for highly efficient attention calculations while preserving critical 3D spatial information necessary for aerodynamic analysis.

**Space-filling Curves.** To maintain spatial proximity in serialized sequences, we employ space-filling curves, specifically Morton (Z-order) and Hilbert curves to serialize the vehicle's surface geometry. After voxelizing the vehicle surface with grid size $g$, each voxel center $\mathbf{v} = (x, y, z) \in \mathbb{Z}^3$ is mapped into a compact 64-bit Morton or Hilbert code $\phi(\mathbf{v}) \in \mathbb{Z}$. Sorting the surface points based on these codes implicitly encodes spatial locality, facilitating effective capture of aerodynamic features such as local pressure distributions and detailed wall shear stress patterns.

**Multi-pattern Rolling.** Using a single serialization pattern may bias the model's attention toward particular directional contexts, potentially limiting its effectiveness in modeling the diverse spatial relationships inherent in aerodynamic phenomena. To mitigate this, we precompute multiple serialization patterns, including Morton, Hilbert, and their axis-permuted variants—and cyclically rotate through these patterns at each encoder layer. This multi-pattern strategy enhances the model's ability to capture varied aerodynamic contexts, ensuring robust modeling of both global shape-dependent effects and localized aerodynamic features.

**Patch Grouping.** Following serialization, the structured sequence is segmented into non-overlapping patches, each of size $M$. Within these localized patches, we apply Serialized Patch Attention (SPA). This design eliminates the computational burden of repeated KNN searches and pairwise relative positional encoding (RPE), significantly reducing complexity. By simplifying computations in this manner, our model effectively captures essential fine-grained aerodynamic details across extensive automotive surfaces, ensuring both accuracy and computational efficiency in large-scale aerodynamic predictions.

### 3.3 HIERARCHICAL MULTI-SCALE ENCODER

**Motivation.** Real-world aerodynamic phenomena span multiple scales, from global structures such as wakes and vortices to fine-grained details like localized flow separations around mirrors or bumpers. Accurately modeling these phenomena requires a multi-scale representation that captures both large-scale interactions and detailed local features. Our hierarchical multi-scale encoder progressively extracts global context first and then refines local aerodynamic details, effectively addressing the complex, scale-dependent nature of aerodynamic flow fields.

**Encoder Structure.** The encoder consists of $N$ Transformer blocks, arranged into $S$ sequential stages. Each block operates on a point set $\mathbf{P}^{(k)}$, transforming input features into richer representations:

$$\mathbf{P}^{(k)} = \texttt{Block}^{(k)}(\mathbf{P}^{(k-1)}), \quad k = 1, \ldots, N. \tag{2}$$

At stage transitions, we down-sample the points by a factor of $1 : 2$ using down sampling and simultaneously double the feature channel width. This hierarchical design enables efficient aggregation of both global and local aerodynamic features across different scales.

**Block Design.** Each block contains three streamlined components: (i) Geometry-aware Position Encoding (GPE) that embeds local geometric context, (ii) Serialized Patch Attention (SPA) that efficiently captures long-range dependencies, (iii) a point-wise MLP for feature refinement. Residual connections and layer normalization further enhance model stability and performance, enabling the block to deliver coherent global features and precise local details for downstream predictions.

### 3.4 GEOMETRY-AWARE POSITION ENCODING (GPE)

**Motivation.** Local geometry—such as surface curvature and edges—strongly influences aerodynamic behavior. Standard positional encodings are insufficient for capturing these nuanced features on irregular surfaces. To address this, we introduce Geometry-aware Position Encoding (GPE), embedding local geometric context through sparse voxel-based 3D convolutions.

**Implementation.** In each Transformer block, incoming features first pass through a sparse 3D convolutional layer, capturing local geometric details such as curvature and sharp edges. A subsequent linear projection and LayerNorm layer stabilize training. GPE enhances the model's sensitivity to local geometric variations, significantly improving prediction quality for both global and local aerodynamic metrics.

### 3.5 SERIALIZED PATCH ATTENTION (SPA)

**Motivation.** Directly applying conventional self-attention to large-scale vehicle meshes (over $10^5$ points) incurs prohibitive computational costs ($\mathcal{O}(N^2)$). Traditional down-sampling methods sacrifice critical local details, while convolutional approaches struggle to capture the long-range dependencies necessary for accurately modeling aerodynamic phenomena. To overcome these challenges, we propose Serialized Patch Attention (SPA), a highly efficient attention mechanism ($\mathcal{O}(NM)$, $M \ll N$) designed specifically for structured, serialized patches.

**Patch Formation.** We partition the serialized point cloud into non-overlapping geodesic patches of size $M$. Each patch preserves intrinsic distances within a local neighborhood, converting an unstructured 3D surface into structured sequences suitable for efficient computation on GPUs.

**Attention Computation.** Within each patch, SPA computes standard multi-head attention:

$$\text{Attn}(\mathbf{F}) = \text{softmax}\left(\frac{\mathbf{QK}^\top}{\sqrt{d_h}}\right)\mathbf{V}, \tag{3}$$

where $\mathbf{Q}, \mathbf{K}, \mathbf{V}$ are linear transformations of patch features. This mechanism efficiently captures detailed local interactions and longer-range aerodynamic dependencies.

### 3.6 PHYSICALLY STRATIFIED MIXTURE-OF-EXPERTS

**Motivation and Challenges.** Our goal is to simultaneously predict surface pressure, wall shear stress, and the drag coefficient in a single forward pass. This unified model promises reduced computational resources, improved data efficiency, and guaranteed physical consistency between local

and global predictions. However, aerodynamic tasks pose specific challenges: (i) loss-scale mismatch between local fields and global metrics, (ii) multi-scale physical interactions from millimeter-scale boundary layers to meter-scale wakes, and (iii) diverse geometric shapes across vehicle types (sedans, SUVs).

**Phys$^2$MoE Architecture.** To address these, we introduce Phys$^2$MoE, a specialized mixture-of-experts architecture featuring: **1. Dual-depth experts.** Experts with varying depths, where shallow experts efficiently capture global flow features such as wake structures, while deeper experts specialize in resolving fine-scale boundary-layer details critical to aerodynamic accuracy. **2. Stratified parameter sharing.** Instead of "all shared" or "all private", Phys$^2$MoE provides $S$ shared experts for geometry-invariant primitives and $K$ task-specific routed experts for pressure, shear, or drag. Shape-dependent tokens (sedan versus SUV) therefore activate different mixtures, suppressing negative transfer across geometry domains. **3. Physics-aware gating.** Tokens are adaptively routed to experts based on physical relevance, effectively balancing different loss scales and minimizing inter-task interference.

At the end of the encoder we attach expert module. The pool is divided into

$$\underbrace{\mathcal{S}_{\mathrm{sh}},\ \mathcal{S}_{\mathrm{dp}}}_{\text{shared}} \quad \cup \quad \underbrace{\mathcal{P}^t_{\mathrm{sh}},\ \mathcal{P}^t_{\mathrm{dp}}}_{\text{private for task } t},$$

where sh / dp denote **shallow** and **deep** experts, respectively. Every task $t \in \{P, WSS, C_d\}$ has a sparse gate $g_t : \mathbb{R}^d \to \Delta^{|\mathcal{S}|+|\mathcal{P}^t|-1}$ that scores *all* shared experts and its own private ones (Equation 4).

$$\widetilde{\mathbf{H}}_t = \sum_{e \in \mathrm{Top}_k} g^{\mathrm{sh}}_{t,e}\, \mathcal{E}^{\mathrm{sh}}_e(\mathbf{H}) + \sum_{e \in \mathrm{Top}_k} g^{\mathrm{dp}}_{t,e}\, \mathcal{E}^{\mathrm{dp}}_e(\mathbf{H}), \tag{4}$$

where the first sum ranges over $\mathcal{S}_{\mathrm{sh}} \cup \mathcal{P}^t_{\mathrm{sh}}$ (shallow experts), the second over $\mathcal{S}_{\mathrm{dp}} \cup \mathcal{P}^t_{\mathrm{dp}}$ (deep experts), and each term keeps the top–$k$ weights within its depth. Shallow experts supply coarse wake features for $C_d$, whereas deep experts resolve near-wall physics for pressure and wall shear stress. The aggregated token set $\widetilde{\mathbf{H}}_t$ is then passed to the corresponding decoder (local fields) or global head (drag).

## 3.7 TASK HEADS

**Local Task Head.** The Local Task Head is responsible for reconstructing detailed flow fields such as surface pressure and wall shear stress. The structure of the Local Task Head mirrors the encoder, but instead of down-sampling, it performs upsampling to recover high-resolution features from the encoder. The output is processed through decoder blocks, applying Grid UnPooling and Serialized Patch Attention (SPA) to refine local features. This allows the model to capture fine-grained details in the flow field, ensuring accurate predictions of pressure and shear stress.

**Global Task Head.** The Global Task Head predicts the overall drag coefficient ($C_d$). To comprehensively capture vehicle shape context, we apply global pooling at multiple encoder scales. These pooled features are concatenated and processed by a regression MLP, ensuring accurate prediction of the global aerodynamic metric.

**Learning objective.** We employ a simple but robust weighted sum of *relative* mean-squared errors (RMSE):

$$\mathcal{L} = \lambda_p\, \mathrm{RMSE}_p + \lambda_\tau\, \mathrm{RMSE}_{\tau_w} + \lambda_d\, \mathrm{RMSE}_{C_d},$$

where $\lambda_p$, $\lambda_\tau$ and $\lambda_d$ are the weighting factors for the pressure, wall shear stress, and drag coefficient, respectively.

## 4 EXPERIMENTS

We conduct extensive experiments to evaluate the performance of UniAero on three industrial datasets, focusing on both global and local aerodynamic tasks.

**Datasets.** We evaluate our model using three industrial datasets: DrivAerNet++, DrivAerNet, and DrivAerML, which cover various vehicle shapes (Estateback, Fastback, Notchback), point resolutions (420k to 8.2M polygons). Table 1 summarizes key

Table 1: Summary of the datasets for evaluation.

| Dataset | #Cars | #Train/Val/Test | #Point |
|---|---|---|---|
| DrivAerNet++ | 7673 | 5361/1148/1154 | 420k–2.2M |
| DrivAerNet | 3760 | 2632/562/566 | 420k |
| DrivAerML | 483 | 383/50/50 | 8.2M |

statistics. To ensure fair comparisons with baseline models, we perform separate training for the global task (drag coefficient prediction) and local task (surface flow field reconstruction), keeping the same data splits across both tasks. Specifically, we partition the training set from one subset while testing on different subsets or datasets.

**Evaluation Metrics.** We evaluate the model's performance using metrics: **Relative $L_2$ Error**: This metric measures the percentage error relative to the ground truth and is defined as:

$$\text{Relative } L_2 \text{ Error} = \frac{\sum_{i=1}^{M} \left(Y_i - \hat{Y}_i\right)^2}{\sum_{i=1}^{M} (Y_i)^2},$$

where $Y_i$ is the true value, $\hat{Y}_i$ is the predicted value and $M$ denotes the number of samples/points.

**Baselines.** We compare UniAero with more than 10 baselines, including state-of-the-art point-based 3D deep learning methods such as PointNet (Qi et al., 2017a), PointNet++ (Qi et al., 2017b), PointTransformer (Zhao et al., 2021),SGFormer (Wu et al., 2023b) ,PointGPT (Chen et al., 2024), Mamba3D (Han et al., 2024) and MeshGraphNet (Pfaff et al., 2021a), as well as neural operator methods like GNOT (Hao et al., 2023), GINO (Li et al., 2023c), DragSolver (Liu & Chen, 2025), AeroGTO (Liu et al., 2025), and Transolver (Wu et al., 2024a). To enable fair comparisons, we adapt local-field-only models by adding a global pooling and fully-connected layer to predict global aerodynamic metrics ($C_d$). Unlike these adapted models, UniAero inherently performs joint modeling of global and local aerodynamic quantities.

**Implementations.** To ensure fair comparisons, we employ a consistent hierarchical encoder backbone comprising 3–5 stages, reducing spatial resolution by half at each stage while increasing feature channels from 32 to 512 and patch size from 512 to 2048 points. For input preparation, we apply a random sampling strategy to uniformly sample approximately 60k points from each vehicle surface. In the Phys²MoE module, we configure a total of 10 experts, including 4 shallow experts (depth of 2 hidden layers) and 6 deep experts (depth of 5 hidden layers). Specifically, 2 shallow and 2 deep experts are shared across tasks, while the remaining 6 experts are task-specific and adaptively routed. All experts have a hidden dimension of 256 channels. For multi-task optimization, we set the loss weights to prioritize local flow fields due to their greater optimization difficulty relative to global drag prediction. Specifically, we set the learning objective weights as $\lambda_p = 1$, $\lambda_\tau = 1$, and $\lambda_d = 0.1$. We optimize the model using the AdamW optimizer (Loshchilov, 2019) with a base learning rate of $1 \times 10^{-3}$, a batch size of 8, and train for 50 epochs with a cosine learning rate decay schedule. Unless otherwise specified, all experiments are conducted on NVIDIA A800 GPUs, and these hyperparameters remain consistent across all experiments. Detailed implementation settings are provided in Appendix C.

## 4.1 GLOBAL TASK PERFORMANCE

We evaluate UniAero on the global task of drag coefficient prediction ($C_d$) across three datasets: DrivAerNet, DrivAerNet++, and DrivAerML. As shown in Table 2, UniAero consistently achieves state-of-the-art results, outperforming point-based networks and advanced neural operators. Compared to the strongest baseline, DragSolver, UniAero achieves relative $L_2$ errors of 0.0007 (DrivAerNet), 0.0006 (DrivAerNet++), and 0.0019 (DrivAerML), reducing errors by up to 14.28%. Crucially, unlike baseline models adapted via global pooling and fully-connected layers, UniAero explicitly leverages the inherent coupling between global and local aerodynamic phenomena through joint modeling. This unified approach enables UniAero to capture richer aerodynamic insights, directly yielding more accurate and robust predictions without additional post-processing.

Table 2: Performance comparison on three datasets for both global drag coefficient $C_d$ and local pressure (P) and wall shear stress (WSS) field prediction. Relative $L_2$ errors are reported for all tasks. The best results are highlighted in **bold**, and the second-best results are underlined. "Promotion" denotes the relative improvement of our best-performing model (**UniAero**) compared to the best-performing baseline model, calculated as $1 - \frac{\text{Our best model's error}}{\text{Best competitor's error}}$. "UniAero-single" represents models trained separately for each task, while "UniAero" denotes the joint training across all three tasks. "/" indicates that the model is not applicable for the corresponding task. Models marked with (*) are adapted to predict $C_d$ by adding global pooling layers and fully-connected layers.

| Model | DrivAerNet | | | DrivAerNet++ | | | DrivAerML | | |
|---|---|---|---|---|---|---|---|---|---|
| | $C_d\downarrow$ | P$\downarrow$ | WSS$\downarrow$ | $C_d\downarrow$ | P$\downarrow$ | WSS$\downarrow$ | $C_d\downarrow$ | P$\downarrow$ | WSS$\downarrow$ |
| PointNet (Qi et al., 2017a) | 0.0062 | / | / | 0.0079 | / | / | 0.0101 | / | / |
| PointNet++ (Qi et al., 2017b) | 0.0046 | / | / | 0.0064 | / | / | 0.0093 | / | / |
| PointTransformer (Zhao et al., 2021) | 0.0024 | / | / | 0.0031 | / | / | 0.0065 | / | / |
| PointGPT (Chen et al., 2024) | 0.0023 | / | / | 0.0028 | / | / | 0.0056 | / | / |
| DragSolver (Liu & Chen, 2025) | 0.0008 | / | / | 0.0007 | / | / | 0.0021 | / | / |
| SGFormer (Wu et al., 2023b)* | 0.0024 | 0.2215 | 0.2528 | 0.0026 | 0.2065 | 0.2426 | 0.0042 | 0.2360 | 0.2650 |
| MeshGraphNet (Pfaff et al., 2021a)* | 0.0064 | 0.2508 | 0.2834 | 0.0041 | 0.2429 | 0.2790 | 0.0092 | 0.2559 | 0.2642 |
| RegDGCNN (Elrefaie et al., 2024b)* | 0.0030 | 0.2845 | 0.3718 | 0.0059 | 0.2705 | 0.3445 | 0.0078 | 0.2829 | 0.3669 |
| GINO (Li et al., 2023c)* | 0.0286 | 0.2360 | 0.2679 | 0.0316 | 0.2359 | 0.2645 | 0.0387 | 0.2365 | 0.2611 |
| Transolver (Wu et al., 2024a)* | 0.0180 | 0.2395 | 0.2745 | 0.0223 | 0.2320 | 0.2610 | 0.0342 | 0.2529 | 0.2714 |
| GNOT (Hao et al., 2023)* | 0.0036 | 0.2536 | 0.2786 | 0.0028 | 0.2479 | 0.2643 | 0.0061 | 0.2682 | 0.2845 |
| AeroGTO (Liu et al., 2025)* | 0.0104 | 0.2375 | 0.2556 | 0.0094 | 0.2132 | 0.2315 | 0.0121 | 0.2373 | 0.2582 |
| **UniAero-single (Ours)** | 0.0008 | **0.1645** | 0.2105 | **0.0007** | 0.1705 | 0.2062 | 0.0022 | 0.2145 | 0.2411 |
| **UniAero (Ours)** | **0.0007** | 0.1685 | **0.2045** | **0.0006** | **0.1665** | **0.2031** | **0.0019** | **0.2035** | **0.2331** |
| Relative Promotion | 12.50% | 25.73% | 19.10% | 14.28% | 19.30% | 12.26% | 9.52% | 13.77% | 9.72% |

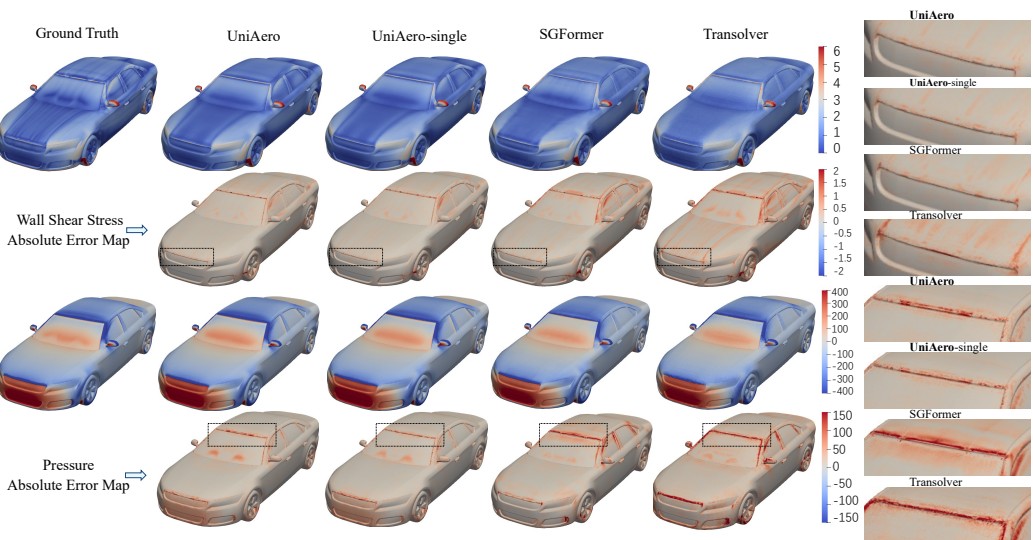

Figure 3: Columns: ground truth, UniAero (multi-task, Phys$^2$MoE), UniAero-single, SGFormer, Transolver. Rows: (1) wall shear magnitude, (2) wall shear absolute error, (3) surface pressure, (4) pressure absolute error (red = higher error). Right: aligned $2\times$ zooms on fixed ROIs for each method (top: WSS error, bottom: pressure error). UniAero yields lower and more spatially uniform errors, notably near the A-pillar and windshield. Shared color scales; fixed camera/lighting.

## 4.2 LOCAL TASK PERFORMANCE

For local surface flow prediction, we evaluate UniAero on the per-point regression of pressure and wall shear stress (WSS). As summarized in Table 2, UniAero also leads on all datasets, on DrivAer-Net++, errors reach 0.1665 (pressure) and 0.2031 (WSS), outperforming SGFormer/AeroGTO. Notably, even compared with our own UniAero-single variant (trained separately for each task), joint training improves consistency and reduces total error. This highlights UniAero's ability to capture multi-scale interactions and shared inductive biases between pressure and shear stress.

### 4.3 VISUAL ANALYSIS

Figure 3 provides a side-by-side comparison on an unseen sedan. UniAero (ours) reproduces both the high-shear ribbons along the wheel arches and the low-pressure footprints on the bonnet with near-CFD fidelity, whereas SGFormer underestimates the front-fender separation and Transolver smears the wake-induced suction peak. The superiority is most evident in the absolute error maps: (i) For wall-shear stress, UniAero exhibits only faint, evenly distributed residuals, whereas the baselines show pronounced streaks of high error along the A-pillar and rear diffuser. (ii) For pressure, our model largely suppresses the broad over-prediction band on the windshield that remains conspicuous in the single-task variant, underscoring the benefit of cross-task supervision provided by Phys$^2$MoE.

### 4.4 WHY PHYS$^2$MOE HELPS

Table 3 evaluates different multi-task learning strategies on DRIVAERNET++. **Single-task UniAero** achieves strong performance by training each task independently, however, this requires maintaining three separate models and fails to leverage inter-task structure. **Phys$^2$MoE (ours)** closes all gaps and achieves the best results on all three tasks: 0.1665 (pressure), 0.2031 (WSS), and $R^2 = 0.9734$. This represents a 38.9% reduction in pressure error and 31.5% in WSS error over naïve MTL, along with a 4.2% improvement in

Table 3: Evaluation on DRIVAERNET++. We compare the single-task UniAero, a naïve multi-task version without MoE, a vanilla MoE variant, and our Phys$^2$MoE. For pressure and wall-shear stress we report relative $L_2$ error; for $C_d$ we report $R^2$ (higher ↑ is better).

| Method | Pressure Rel. $L_2 \downarrow$ | WSS Rel. $L_2 \downarrow$ | $C_d$ $R^2 \uparrow$ |
|---|---|---|---|
| UniAero (single) | 0.1705 | 0.2062 | 0.9623 |
| UniAero (w/o MoE) | 0.2731 | 0.2965 | 0.9341 |
| UniAero (w/ MoE) | 0.1938 | 0.2218 | 0.9679 |
| UniAero (w/ Phys$^2$MoE) | **0.1665** | **0.2031** | **0.9734** |

drag prediction. These results confirm that Phys$^2$MoE effectively addresses gradient conflict and task heterogeneity through stratified expert routing, using shallow shared experts to model global wake structures for drag, and deep task-specific experts to capture localized boundary-layer physics, resulting in more accurate aerodynamic predictions.

### 4.5 TRAINING AND INFERENCE EFFICIENCY

We now compare the efficiency of UniAero against state-of-the-art 3D point-based methods under the same hardware environment. As highlighted in previous works, KNN-based approaches (e.g., PointGPT (Chen et al., 2024)) incur high computational costs due to repeated neighbor graph con-

Table 4: Training/inference time (per epoch, per sample) for UniAero and baselines.

| Method | PointGPT | SGFormer | UniAero |
|---|---|---|---|
| Train (s/epoch) | 1680 | 1010 | **540** |
| Infer (s/sample) | 0.97 | 0.95 | **0.93** |

struction and attention computation. To overcome this, UniAero adopts a sparse convolutional backbone (Contributors, 2022) combined with FlashAttention (Dao, 2024), significantly reducing computational complexity by eliminating costly KNN operations. Table 4 demonstrates that UniAero achieves 3× faster training time compared to traditional point-based methods.

## 5 CONCLUSION

We introduce UniAero, a unified Transformer framework for both global and local aerodynamic predictions. It integrates multi-scale encoding, serialized patch attention, geometry-aware position encoding, and physically stratified Mixture-of-Experts. Experiments show that UniAero reduces $C_d$ prediction error by 12%, improves local field accuracy by 16%, and achieves inference times of 1 s per vehicle, significantly faster than CFD. This demonstrates UniAero's potential for real-time aerodynamic design.

ETHICS STATEMENT

This work uses publicly available aerodynamic datasets (DrivAerNet/++, DrivAerML) and involves no human subjects, personal data, or sensitive content; IRB approval was not required. We comply with licenses and cite all sources. Potential misuse: optimizing drag without safety/regulatory constraints, hence we position UniAero as a surrogate within established engineering workflows. All authors have read and adhere to the ICLR Code of Ethics.

REPRODUCIBILITY STATEMENT

We provide materials to facilitate reproduction of our results:

**Code and configs:** An anonymized repository (supplementary) with data loaders, preprocessing scripts, Phys$^2$MoE/Serialized Patch Attention/GPE modules, training/evaluation scripts, and experiment configs.

**Data and splits:** Datasets and preprocessing are described in the main text; exact train/val/test splits appear in Tables 1.

**Hyperparameters:** Optimizer, learning rates/schedules, loss weights ($\lambda_p, \lambda_\tau, \lambda_d$), batch sizes, and epochs are specified in Implementation Details and config files.

**Determinism:** Random seeds are set where supported; any non-deterministic kernels are noted.

**Metrics and evaluation:** Definitions for Relative $L_2$ (fields) and $R^2$ ($C_d$) are given in Evaluation Metrics; scripts to compute them are provided.

**Environment:** An environment file (conda/pip) plus CUDA/driver and GPU details are included. These materials enable reproduction of main results, ablations, and qualitative visualizations with minimal effort.

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

# A DATASET DESCRIPTIONS

In this appendix, we provide detailed descriptions of the datasets used in our experiments: **DrivAer-Net**, **DrivAerNet++**, and **DrivAerML**. These datasets specifically cater to automotive aerodynamic analysis, capturing various vehicle geometries, aerodynamic features, and resolution scales, thus forming a comprehensive benchmark for both global and local aerodynamic predictions.

## A.1 DRIVAERNET

DrivAerNet (Elrefaie et al., 2024a) is a dataset designed to support aerodynamic predictions, particularly focusing on accurately capturing the global drag coefficient ($C_d$) of automotive vehicles. The dataset comprises:

- Vehicle Types: Three distinct vehicle shapes are included: Estateback, Fastback, and Notchback.
- Data Scale: The dataset contains 3,760 vehicle geometries, each represented by approximately 420k mesh vertices.
- Data Splits: For our experiments, we follow the standard split: 2,632 vehicles for training, 562 vehicles for validation, and 566 vehicles for testing.
- Aerodynamic Quantities: High-fidelity computational fluid dynamics (CFD) simulation results provide the ground-truth aerodynamic data, particularly the drag coefficient ($C_d$).

DrivAerNet is notable for its structured geometric diversity and fidelity, making it a suitable benchmark for evaluating surrogate models targeting global aerodynamic predictions.

## A.2 DRIVAERNET++

DrivAerNet++ (Elrefaie et al., 2024b) extends DrivAerNet by significantly increasing the diversity and complexity of vehicle geometries and flow conditions. Key characteristics include:

- Vehicle Types: Expanded variety of vehicle geometries, including Estateback, Fastback, and Notchback, with additional configurations to cover broader aerodynamic phenomena.
- Data Scale: The dataset consists of 7,673 vehicle geometries, with mesh resolutions ranging from 420k to 2.2M vertices per vehicle, enabling evaluation at multiple spatial scales.
- Data Splits: It is partitioned into 5,361 vehicles for training, 1,148 vehicles for validation, and 1,154 vehicles for testing.
- Aerodynamic Quantities: Ground-truth CFD results include detailed local aerodynamic flow fields such as surface pressure and wall shear stress, along with global drag coefficients.

The DrivAerNet++ dataset is particularly suited for comprehensive evaluations of multi-task surrogate models, providing benchmarks for both global metrics and detailed local flow field reconstructions.

## A.3 DRIVAERML

DrivAerML (Ashton et al., 2024) represents the most detailed and high-resolution dataset among the three, specifically created to push the boundaries of geometric resolution and aerodynamic feature detail:

- Vehicle Types: The dataset covers highly detailed automotive geometries, focusing on complex aerodynamic scenarios often encountered in advanced design processes.
- Data Scale: It includes 483 vehicles, each represented by approximately 8.2 million mesh vertices, offering an unprecedented level of detail.
- Data Splits: For consistency, we follow the standard partition of 383 vehicles for training, 50 vehicles for validation, and 50 vehicles for testing.

- Aerodynamic Quantities: CFD simulation results in DrivAerML provide exceptionally high-fidelity local aerodynamic quantities such as detailed pressure fields and precise wall shear stress distributions, in addition to global drag metrics.

DrivAerML's high resolution and extensive detail make it ideal for evaluating models' ability to handle large-scale data and accurately predict intricate aerodynamic phenomena, pushing surrogate modeling towards near-CFD precision.

Collectively, these three datasets provide a robust foundation for rigorous evaluation and benchmarking of deep learning-based aerodynamic prediction methods, covering a broad spectrum of geometric complexities, aerodynamic features, and scales.

# B    DETAILED MODEL PARAMETERS

**UniAero Model Configuration**

## B.1    OVERALL ARCHITECTURE

Our model architecture follows the hierarchical Transformer structure with serialized patch attention and utilizes Physically Stratified Mixture-of-Experts at a specified encoder stage. Below are detailed hyperparameters:Input Channels=3, Serialization Order: ["z", "hilbert"], and Encoder Stages = 5, Decoder Stages = 4, Embedding Channels = 32.

## B.2    ENCODER CONFIGURATION

The encoder is organized into five stages with the following parameters in Table 5:

| Stage | Stride | Blocks | Channels | Heads | Patch Size |
|-------|--------|--------|----------|-------|------------|
| 1 | - | 2 | 32 | 2 | 512 |
| 2 | 2 | 2 | 64 | 4 | 512 |
| 3 | 2 | 2 | 128 | 8 | 512 |
| 4 | 2 | 4 | 256 | 16 | 512 |
| 5 | 2 | 2 | 256 | 16 | 512 |

Table 5: Encoder stage-wise configurations.

## B.3    DECODER CONFIGURATION

The decoder mirrors the encoder structure and has four stages in Table 6:

| Stage | Blocks | Channels | Heads | Patch Size |
|-------|--------|----------|-------|------------|
| 1 | 2 | 128 | 8 | 512 |
| 2 | 2 | 64 | 4 | 512 |
| 3 | 2 | 32 | 2 | 512 |
| 4 | 2 | 32 | 2 | 512 |

Table 6: Decoder stage-wise configurations.

## B.4    PHYSICALLY STRATIFIED MIXTURE-OF-EXPERTS CONFIGURATION

The MOE module is integrated after encoder stage 3 (zero-indexed), with the following parameters:

- MoE Position: After encoder stage 3
- Number of Experts: 10
- Number of Tasks: 3 (Pressure, Wall Shear Stress, Drag Coefficient)

- Expert Network:

  - Type: MLP-based with optional Flash Attention
  - Hidden Layer Dimensionality Ratio: 0.5 (relative to input channels)
  - Depth: 3 or 5 layers per expert
  - Attention: Flash Attention variant available

- Gating Network: Softmax-based task-specific gating

## B.5 TASK HEADS

The model has separate heads for each task:

- Pressure Head: Linear layer, input channels = 32, output = 1

- Wall Shear Stress Head: Linear layer, input channels = 32, output = 3 (vector)

- Drag Coefficient ($C_d$) Head: Fully connected layers

  - Input Channels: 512 (concatenation of global mean and max pooling from the final encoder)
  - Hidden layers: [128, 64]
  - Activation: ReLU with BatchNorm
  - Dropout: 0.5
  - Output Channels: 1 (scalar)

## B.6 ADDITIONAL HYPERPARAMETERS

- Drop Path Rate: 0.3

- Normalization: Layer Normalization (LayerNorm), Batch Normalization (BatchNorm)

- Activation: GELU

- Projection Dropout: 0.0

- Attention Dropout: 0.0

- Use Flash Attention: True

- Relative Positional Encoding (RPE): Disabled

## C IMPLEMENTATION DETAILS

### C.1 TRAINING CONFIGURATION

We optimize our UniAero model using the AdamW optimizer (Loshchilov, 2019) with a base learning rate of $1 \times 10^{-3}$. The training batch size is set to 8, and the total training duration is 50 epochs, with a cosine learning rate decay schedule applied to smoothly decrease the learning rate towards the end of training. To handle varying vehicle surface resolutions efficiently, we randomly sample approximately 60k points uniformly from each vehicle surface before feeding them into the model.

The Phys$^2$MoE module within UniAero employs 10 expert networks: 4 shallow experts (each with 2 hidden layers) and 6 deep experts (each with 5 hidden layers). Among these, 2 shallow and 2 deep experts are shared across all tasks, while the remaining 6 experts are task-specific and adaptively routed through a gating mechanism. Each expert network has a hidden dimension of 256 channels, and GELU activation functions are used throughout the network.

We conduct our training on 4 NVIDIA A800 GPUs, with each GPU having 64 GB of memory to ensure adequate resources for handling large-scale point cloud data and batch sizes.

## C.2 Loss Function

Our model is jointly optimized for global (drag coefficient $C_d$ prediction) and local (pressure $P$ and wall shear stress $\tau$ fields) aerodynamic tasks. The total loss $\mathcal{L}$ combines task-specific losses weighted according to their relative optimization difficulty:

$$\mathcal{L} = \lambda_p \mathcal{L}_p + \lambda_\tau \mathcal{L}_\tau + \lambda_d \mathcal{L}_d, \tag{5}$$

where $\mathcal{L}_p$, $\mathcal{L}_\tau$, and $\mathcal{L}_d$ are the mean squared errors (MSE) losses computed between predictions and ground truths for pressure, wall shear stress, and drag coefficient, respectively. Specifically, we use:

$$\lambda_p = 1, \tag{6}$$
$$\lambda_\tau = 1, \tag{7}$$
$$\lambda_d = 0.1. \tag{8}$$

This weighting scheme prioritizes local flow field predictions due to their higher complexity and difficulty in optimization compared to the scalar drag prediction.

## C.3 Evaluation Metrics

In addition to the Relative $L_2$ Error metric presented in the main experiments, we also use the Coefficient of Determination ($R^2$) to evaluate the global drag prediction task. This metric measures the proportion of variance in the true drag coefficients explained by the model:

$$R^2 = 1 - \frac{\sum_{i=1}^{M} \left( C_{d,i} - \hat{C}_{d,i} \right)^2}{\sum_{i=1}^{M} \left( C_{d,i} - \overline{C}_d \right)^2}, \tag{9}$$

where $C_{d,i}$ is the ground-truth drag coefficient, $\hat{C}_{d,i}$ is the predicted drag coefficient, and $\overline{C}_d$ is the mean of the ground-truth drag coefficients. An $R^2$ value closer to 1 indicates better predictive performance.

## C.4 Comparison Methods

We compare UniAero against various advanced baseline methods spanning two main categories:

- **Point-based 3D deep learning methods**, including PointNet (Qi et al., 2017a), Point-Net++ (Qi et al., 2017b), PointTransformer (Zhao et al., 2021), PointGPT (Chen et al., 2024), SGFormer (Wu et al., 2023b), and Mamba3D (Han et al., 2024).
- **Neural operator and graph-based methods**, including MeshGraphNet (Pfaff et al., 2021a), GNOT (Hao et al., 2023), GINO (Li et al., 2023c), DragSolver (Liu & Chen, 2025), AeroGTO (Liu et al., 2025), RegDGCNN (Elrefaie et al., 2024b), and Transolver (Wu et al., 2024a).

These comparisons provide a comprehensive performance evaluation, highlighting our method's advantages in both accuracy and computational efficiency across global and local aerodynamic tasks. To enable baseline methods originally designed solely for local flow-field predictions to also predict global aerodynamic metrics ($C_d$), we adapted them by appending a global pooling layer followed by fully-connected layers. Specifically, we first perform a global pooling operation (concatenating mean and max pooling results), resulting in a feature vector of twice the original channel dimension. This pooled feature is then passed through a fully-connected module consisting of three linear layers interleaved with Batch Normalization, ReLU activations, and dropout regularization:

- Layer 1: Linear (input dimension: 2 × feature channels, output dimension: 128), Batch-Norm1d, ReLU activation, Dropout (probability=0.5)
- Layer 2: Linear (128, 64), BatchNorm1d, ReLU activation
- Layer 3: Linear (64, 1), outputting the scalar drag coefficient prediction.

These modifications ensure a fair and consistent evaluation against our UniAero model, which inherently supports joint prediction of global and local aerodynamic properties.

# D MODEL PERFORMANCE SENSITIVITY ANALYSIS

We conduct sensitivity experiments on the DRIVAERNET++ dataset to investigate the impact of loss weighting parameters ($\lambda_p$, $\lambda_\tau$, and $\lambda_d$) on the multi-task performance of UniAero. Table 7 summarizes the results across different weighting schemes, focusing on their influence on both global and local aerodynamic predictions.

Our analysis reveals that increasing the drag coefficient loss weight to equal the local field weights ($\lambda_p = 1$, $\lambda_\tau = 1$, $\lambda_d = 1$) slightly improves local predictions (pressure and WSS) but results in a notable decline in the accuracy of drag coefficient predictions ($R^2 = 0.9650$). Conversely, reducing the weight of the wall shear stress loss ($\lambda_p = 1$, $\lambda_\tau = 0.5$, $\lambda_d = 0.1$) achieves marginally better accuracy in both drag coefficient and pressure predictions, demonstrating the beneficial trade-off when emphasizing specific tasks.

The baseline configuration ($\lambda_p = 1$, $\lambda_\tau = 1$, $\lambda_d = 0.1$) achieves the best balance overall, providing optimal performance across all tasks with a notably high $R^2$ score of 0.9734 for the drag coefficient and low relative $L_2$ errors for both pressure (0.1665) and wall shear stress (0.2031). This configuration underscores the importance of careful weight selection to achieve robust multi-task performance, effectively balancing local flow field accuracy with global aerodynamic metrics.

Table 7: Sensitivity Analysis of Model Performance under Varying Loss Weights. Relative $L_2$ errors are reported for pressure (P) and wall shear stress (WSS), and $R^2$ score is reported for drag coefficient ($C_d$). The default weights ($\lambda_p = 1$, $\lambda_\tau = 1$, $\lambda_d = 0.1$) are highlighted in bold.

| $\lambda_p$ | $\lambda_\tau$ | $\lambda_d$ | P (Rel. $L_2$)↓ | WSS (Rel. $L_2$)↓ | $C_d$ ($R^2$)↑ |
|---|---|---|---|---|---|
| 1 | 1 | 1 | 0.1687 | 0.2057 | 0.9650 |
| 1 | 0.5 | 0.1 | **0.1646** | 0.2041 | 0.9731 |
| **1** | **1** | **0.1** | 0.1665 | **0.2031** | **0.9734** |
| 1 | 1 | 0.5 | 0.1668 | 0.2037 | 0.9728 |

# E LLM USAGE (LANGUAGE EDITING ONLY).

A large language model was used exclusively for copy-editing (grammar, wording, and minor stylistic tightening). It was **not** involved in problem framing, method or experiment design, data handling or analysis, interpretation of results, or drawing conclusions. All edits were reviewed and approved by the authors.

