# OpenReview forum: "UniAero: A Unified Framework for Global Drag and Local Flow Field Prediction"
_ICLR.cc/2026/Conference — ICLR 2026 Conference Withdrawn Submission_

### Official Review · Reviewer_AWCR · 2025-10-26

**Soundness:** 1
**Presentation:** 2
**Contribution:** 1
**Rating:** 0
**Confidence:** 4

**Summary:**

This paper aims to develop a surrogate model with novel components to predict both the global and local flow properties of car geometries, better handling cross-scale prediction. The car geometries are 2D surfaces in the 3D space. The authors break this 2D surface down into space filling curves that can be passed to transformer-like backbones (employing SPA) and Phys2MoE to derive accurate cross-scale predictions for the global drag coefficient and the local pressure and wall shear stress.

**Strengths:**

1. The paper is well structured.
2. The method seems to work much better than the other baselines.
3. Clear illustration of the model architecture and components. Appealing visual illustration of the car samples and errors.

**Weaknesses:**

The application of this surrogate model is very limited, and the selected relative L2 metric is also problematic, as it may exaggerate performance improvement. Moreover, the claims about the usefulness of different components need to be better proved and analyzed to provide more insights. If no theoretical insights can be provided about these components, the work should at least support its claims with more rigorous and thorough experiments to make up for its limited application. Based on these points I suggest rejecting this paper.

## Specifically:

1. The CFD surrogate model appears restricted to a very narrow class of geometries (automobiles) with fixed flow boundary conditions. The practical significance of this application should be demonstrated, like how important this is to the car industry in their car development, how much this surrogate can expedite their prototyping, how much money can be saved, what the car manufacturers' typical tolerance to the prediction error is, do they always need the prediction only for a fixed flow condition.
2. If the authors want to suggest its potential extrapolation to the other geometries, then the surrogate's generalization capability remains unclear due to the limited car dataset.
    * Its performance may also need to be evaluated on the training set, and reported in the appendix.
    * The shape differences between the training and test sets should be systematically evaluated to understand how the degree of geometric variation affects prediction accuracy. The cars seem to have similar global shapes.
    * The model’s performance on simple unseen geometries (e.g., a sphere) also needs to be examined. If it's not good, this limitation should be highlighted.
3. Due to this narrow application I expect the authors to at least shed some light on the component designs. However, there is no rigorous ablation test conducted to study their unique properties and verify each novel component’s contribution to the performance (except MoE). This significantly undermines the fancy claims made in Section 2 and 3 about these components’ usefulness.
    * I suggest that the authors check the AlphaFold2 paper to understand how to write a high quality paper for a model with many novel components, without making shaky claims.
4. In Table 2, the number of parameters for each model should be reported to verify whether the improvement in accuracy arises from increased model complexity, especially for MoE. If proper, a plot should also be generated to illustrate this.
5. The relative L2 error alone—which is a scalar—is not enough to illustrate how better this model is at capturing cross-scale phenomenon compared to the other models. For instance, how does the model perform in regions with sharp corners and high curvatures? The visual inspection in Fig. 3 over only a few samples is not enough for this. I expect a more systematic, numerical evaluation over all test samples.
   * For instance, a 2D graph Laplacian may be used to roughly evaluate the car surface's nonlinearity at each point, and the local prediction errors can be weighted by these Laplacian coefficients to derive a global metric for this purpose.
6. Additionally, L1 might be a better choice than this L2 (which is actually MSE in this work) due to the high dimensionality of this problem. It's also weird that the authors chose L1 in Fig. 3 but relative error based on MSE in Table 2. I wonder if the relative promotion in Table 2 look appealing because of this choice.
   * **To resolve this confusion, I strongly suggest that the authors use relative absolute error for Table 2**.
   * _Aggarwal, C. C.; Hinneburg, A.; Keim, D. A. On the Surprising Behavior of Distance Metrics in High Dimensional Space. In Database Theory — ICDT 2001; Van den Bussche, J., Vianu, V., Eds.; Lecture Notes in Computer Science; Springer: Berlin, Heidelberg, 2001; pp 420–434. https://doi.org/10.1007/3-540-44503-X_27._

**Questions:**

1. The generation of space filling curves is built on voxelizing the vehicle surface with fixed grid size. If I understand it correctly, this will make the resulting 1-D representation have equal resolution over the entire car surface.
    1. How does this help handle regions with critical geometric features that induce complex flow structures and therefore require higher spatial resolution to represent?
        1. For instance, FEA requires mesh refinement in areas where the solution field exhibits large spatial gradients or curvature.
        2. Because of this limitation, the claims about the cross-scale performance are further undermined.
    2. Is the space-filling curve continuous, or it has occasional discontinuous jumps on the car surface? It would be great if this curve can be illustrated on the car's surface to address this confusion.
        1. If so, how does the discontinuous jumps affect the performance?
2. In Sec. 3.5, how exactly does the patch formulation help overcome the "struggle to capture the long-range dependences..." of convolution? I don't think the information can pass between patches, right?

---

### Official Review · Reviewer_ycQD · 2025-10-29

**Soundness:** 3
**Presentation:** 3
**Contribution:** 2
**Rating:** 4
**Confidence:** 5

**Summary:**

This paper proposes UniAero, a unified transformer-style surrogate model for automotive external aerodynamics. Given only the 3D vehicle surface geometry (high-resolution point cloud / mesh), UniAero predicts in a single forward pass both: a global performance scalar, the drag coefficient ($C_d$), and detailed surface-resolved flow quantities, including surface pressure and wall shear stress (WSS) at each point on the body.

The motivation is that current industrial pipelines rely on computational fluid dynamics (CFD), which can take hours to days per design, and that ML surrogates typically specialize in either global metrics (e.g., ($C_d$)) or local flow reconstruction, but not both. The authors argue that these two targets are physically coupled (local flow separation and near-wall stresses influence global drag) and should be predicted jointly.

**Strengths:**

1. The paper introduces UniAero, which predicts both a global metric $C_d$ and dense per-point surface fields (pressure, wall shear stress) from only the 3D vehicle surface in a single forward pass. This directly targets an industrial bottleneck: CFD-style feedback is slow, and prior ML surrogates typically only predict either drag or local fields, not both jointly. UniAero explicitly treats these quantities as physically coupled, which is novel and valuable for real design iteration.

2. The model combines (i) Serialized Patch Attention, which orders and chunks the car-surface point cloud so attention is computationally tractable while still passing long-range wake information, (ii) a hierarchical geometry-aware encoder with curvature-aware position encoding to capture both global wake structures and sharp local features like A-pillar separation, and (iii) Phys2MoE, a physics-informed mixture-of-experts that separates global/wake experts from near-wall/pressure-shear experts to avoid cross-task interference. These are technically nontrivial advances beyond generic point-cloud transformers or naive multi-task heads.

**Weaknesses:**

1. The paper thoroughly ablates Phys2MoE, showing it outperforms naïve multi-tasking and vanilla MoE, but it does not isolate how much Serialized Patch Attention vs. a simpler local attention, or Geometry-aware Position Encoding vs. plain positional features, each contribute to accuracy, runtime, or memory. A “w/o SPA / w/o GPE” study would make the source of gains clearer.

2. The paper reports training time per epoch and ~1 s inference per car and claims improved efficiency over KNN-heavy baselines, but does not report FLOPs, parameter count, peak training/inference memory, or scaling with point count. That limits how convincingly we can judge deployability in a real aero design loop.

3. The joint loss uses hand-tuned task weights ($\lambda_p,\lambda_\tau,\lambda_d$), and performance is sensitive to those values. This raises concerns about extensibility: as more targets are added (lift, side force, aeroacoustics, etc.), manual λ tuning becomes brittle and harder to reproduce.

4. UniAero relies on voxelizing and serializing very large surface point clouds (hundreds of thousands to millions of points) into multiple Morton/Hilbert-style orderings before inference. For new high-resolution geometries, that preprocessing itself may be nontrivial and could erode the claimed “interactive” turnaround time.

5. Some terminology and structure choices reduce accessibility — e.g. “naïve MTL” appears before “multi-task learning” is clearly defined, and core training details (task weights, downsampling strategy) are scattered rather than surfaced early. In addition, many architectural elements (hierarchical point-cloud transformer backbone, multi-scale decoder heads, Mixture-of-Experts routing) resemble known designs; without a clearer, sharper statement of what is *algorithmically* new vs. what is an effective integration for automotive aerodynamics, readers may perceive the work as a strong systems combination rather than a fundamentally new learning paradigm.

**Questions:**

1. How sensitive is UniAero to geometric classes not present in training? For example, can a model trained mostly on sedan/SUV-like bodies generalize to boxy vans, pickup trucks, or other high-departure geometries without retraining? If you tested any such OOD scenarios, please report both drag error and surface-field error.
If you did not explicitly test extreme geometries, can you instead report cross-dataset generalization (e.g., train on two of the three datasets and evaluate zero-shot on the third) to quantify how well UniAero transfers to unseen car classes / mesh resolutions?

2. Did you verify basic physics consistency between predicted fields and predicted ($C_d$)?

   - If you integrate the predicted surface pressure and wall shear stress over the body to obtain an implied drag force, how close is that to the directly predicted ($C_d$)?
   - If you performed that check, please quantify the mismatch; if not, can you comment on feasibility and whether this is something the model could be encouraged to satisfy?

3. The reported pressure prediction error appears noticeably higher than expected, and higher than what is typically shown in Transolver and AeroGTO for surface pressure reconstruction. Can you explain why? Also, did you try training the baselines (e.g., Transolver or AeroGTO method) in a true multi-task or multi-dataset setting? If so, how do they compare to UniAero?

4. Serialized Patch Attention (SPA) relies on spatial orderings (Morton / Hilbert curves, axis permutations) and rotates these orderings layer by layer.

   * How sensitive is accuracy to the specific ordering strategy used for serialization?
   * Did you compare SPA to graph-based local attention (e.g., KNN neighborhood attention) at similar FLOPs, to confirm that SPA improves not only runtime but also prediction quality?

5. Phys2MoE routes features to “global/wake” experts vs. “near-wall/boundary-layer” experts. How stable is this specialization across random seeds and training runs? Do you consistently observe distinct experts, or does one expert tend to dominate (“mode collapse”)? Qualitative results of routing patterns over the vehicle surface would strengthen the claim that experts align with physically meaningful regions.

---

### Official Review · Reviewer_66yT · 2025-11-01

**Soundness:** 3
**Presentation:** 3
**Contribution:** 2
**Rating:** 4
**Confidence:** 3

**Summary:**

This paper proposes UniAero, a unified deep learning framework for jointly predicting both a global aerodynamic metric (the drag coefficient C_d) and detailed local flow fields (surface pressure and wall shear stress) from 3D car geometries. Unlike prior works that focus on either global or local aerodynamic surrogates independently, UniAero couples these tasks through a single Transformer-based model. Its main innovations include: (i) a Physically Stratified Mixture-of-Experts with scale-sensitive experts and physics-aware gating; (ii) Serialized Patch Attentionfor efficient processing of large automotive meshes while preserving long-range dependencies; and (iii) a hierarchical encoder with Geometry-aware Position Encoding to capture global and local geometric cues. Experiments on three datasets (DrivAerNet, DrivAerNet++, and DrivAerML) show UniAero reduces drag error by up to 14% and improves local flow accuracy by 16%, achieving inference times of 1s per vehicle.

**Strengths:**

The paper tackles the unsolved problem of jointly predicting global and local aerodynamic quantities, explicitly modeling their physical coupling. The model’s architecture is carefully designed for the multi-scale nature of automotive flows. Serialized Patch Attention and GPE are also creative, domain-inspired adaptations that seem to make the model sensitive to fine geometric cues. This is conceptually similar to DragSolver’s focus on multi-scale feature extraction for both global shape and fine details but UniAero extends it to multi-task learning.
UniAero achieves SOTA performance on benchmarks, outperforming DragSolver, SGFormer, and AeroGTO by large margins (12–19% improvement). The qualitative visualizations show more accurate and spatially uniform flow reconstructions. These gains, while modest in absolute terms, indicate that the unified model does not compromise accuracy on either task; in fact, it achieves new state-of-the-art results on combined global/local prediction. The authors evaluate on three datasets, compare to 10+ baselines, and include ablations (e.g., Phys2MoE variants), runtime analysis, and visualization. The experiments seem comprehensive and credible.

**Weaknesses:**

Missing related work: The authors do not cite or compare their work to “Spatially-aware Transformer Operator for Real-Time Aerodynamic Evaluations of Arbitrary Three-Dimensional Vehicles” (Journal of Computational Physics, 2025), which introduces a spatially-aware operator leveraging physics-attention and space-filling curve serialization. There are notable conceptual similarities between that work and UniAero, especially regarding the use of Space-Filling Curves and Serialized Patch Attention for patch grouping. This omission is concerning since both works share overlapping ideas in efficient, spatially structured transformer designs for aerodynamic evaluations.

It is not clear whether the multi-scale modeling in UniAero, while effective, is obviously superior to existing techniques like Dragsolver or AeroGTO.  It seems like a combination of local-patch attention and MoE, rather than a fundamentally new multi-scale paradigm. An expert in CFD might question whether the model truly captures all relevant scales (from millimeter surface details to meter-scale wakes) or if it’s implicitly filtering some due to the patch segmentation. Any evidence that serialization order/patching doesn’t truncate key couplings?

Inconsistent Transolver Results. The reported performance of Transolver on the DrivAerNet++ dataset appears to differ substantially from some other published results, such as those in “AB-UPT: Scaling Neural CFD Surrogates for High-Fidelity Automotive Aerodynamics Simulations via Anchored-Branched Universal Physics Transformers.” arXiv:2502.09692 (2025). Please clarify whether these differences arise from differences in dataset splits, normalization procedures, training hyperparameters, or other experimental configurations. A brief explanation would help ensure fair comparison and reproducibility across studies.

**Questions:**

Apart from some points mentioned above, please look into the following questions:

1. How sensitive are results to patch size/order and expert depth counts?

2. Expand on the physics-aware gating: What signals inform routing decisions, and how does it partition tasks?

3. Please provide a short qualitative analysis of cases where predictions fail or are less accurate. This will help in understanding limitations of the data-driven approach. Are these poor predictions similar across different methods or unique to specific methods (baselines and your results)?

4. Does joint learning trade per-task optimality for consistency? Why does UniAero-single slightly outperform UniAero on some metrics? From an engineering perspective, one must ask: do we need one model to do everything, or would two specialized models (one for drag, one for pressure) suffice in a design pipeline? UniAero’s own results indicate that a single-task drag model (UniAero-single) already gets very low error (R2 around 0.97), and the unified model improves that only slightly.

5. Does Serialized Patch Attention introduce patch boundary artifacts or limit global information flow?

6. What is the physical meaning of the reported relative L_2 errors (e.g., corresponding percentage error in C_d)?

---

### Official Review · Reviewer_vLAN · 2025-11-06

**Soundness:** 2
**Presentation:** 3
**Contribution:** 2
**Rating:** 4
**Confidence:** 4

**Summary:**

The authors propose UniAero, which is a unified framework to jointly predict the drag coefficient $C_d$ and surface fields. The method combines a Physically Stratified MoE (Phys^2MoE) with scale-sensitive experts and physics-aware and physics-aware gating for multi-task, multi-scale learning. They also introduce Serialized Patch Attention to process large meshes efficiently while preserving long-scale interactions. Lastly, they use a hierarchical encoder with Geometry-aware Position Encoding (GPE) to capture shape cues.

Overall, I think the work targets a very high-impact area and is effective. I am concerned about the novelty of the method for ICLR and I think it may be better suited for a more applied journal at its current stage.

**Strengths:**

- Nice real-world ML application to aerodynamics
- Good use of real-world challenging DrivaerML 3D dataset benchmark
- Uniaero is effective at reducing the drag error by 12% and improving accuracy of local-field by 12%
- Nice mix of modeling both local and global interactions.
- Good motivation to the automotive domain
- Challenging problem with multi-scale interactions
- Large number of baselines including the relevant GINO
- Good to see the physical plots of the flow rather than just the metrics
- Nice ablation study in 4.4

**Weaknesses:**

- Authors only predict surface fields instead of the full 3D volume fields
- Very applied paper instead of novel in the methodology
- Missing reference to Chalapathi et al., "Scaling Physics-Informed Hard Constraints with Mixture-of-Experts", ICLR, 2024, which also is an efficient MoE model for PDEs
- Missing reference to Ananthan et al., "Machine learning for road vehicle aerodynamics", SAE Technical Paper, 2024, which tests various methods including MeshGraphNets and U-Net on the DrivAerML dataset.
- I think the sentence on line 074 is too strong stating that GNNs and Transformers can't capture these dependencies. For example, MeshGraphNet (Pfaff et al., ) is a GNN-based method that performs strongly on these types of problems and Transformer methods especially Alkin et al., "AB-UPT for Automotive and Aerospace Applications' (https://arxiv.org/pdf/2510.15808) have also been successful.
- Missing comparisons to state-of-the-art methods, e.g., X-MeshGraphNet and Neural Operator-based DoMino
- Missing reference to effective Transformer models: Alkin et al., "AB-UPT for Automotive and Aerospace Applications' (https://arxiv.org/pdf/2510.15808) and Janny et al., "Eagle: Large-Scale Learning of Turbulent Fluid Dynamics with Mesh Transformers", ICLR 2023.
- The point cloud approach and then KNN is very similar to the approach used in X-MeshGraphNet (https://arxiv.org/abs/2411.17164) and should be cited and compared to.
- Hierarchical structures have also used in past works, e.g., Multi-scale Meshgraphnet (Fortunato et al., https://arxiv.org/abs/2210.00612) which is the foundation for graphcast (Lam et al., Science, https://www.science.org/doi/10.1126/science.adi2336). Both of which should also be cited.
- Note Pfaff et al., is cited twice.
- With three penalty parameters, the loss may not be robust and requires significant HPO similar to PINNs and its challenges (See Krishnapriyan et al., "Characterizing possible failure modes in physics-informed neural networks", https://arxiv.org/abs/2109.01050)
- The method is deterministic - exploring probabilistic approaches may also be interesting follow-ups

**Questions:**

1. Why do the authors not present the full volume prediction? Was it due to computational or memory challenges?
2. What is the effect of the patch size?
3. What do the authors mean in the abstract by "subtle shape cues"?
4. To calculate the global drag coefficient, couldn't simple tabular regression methods even be used, i.e., AutoGluon or SciKitLearn?
5. Please indicate what are the key novelties of the proposed approach
6. What is the purpose of using point clouds instead of inputting the data as a mesh as done in MeshGraphNets? Is it due to scalability or memory issues?
7. How do the authors determine the weights in the loss function? C.2 just gives their values but doesn't discuss how they are determined.
8. Have the authors thought of applying any physical constraints?

---

### Note · Authors · 2025-11-14

I have read and agree with the venue's withdrawal policy on behalf of myself and my co-authors.